# The Expression of the Endocannabinoid Receptors CB2 and GPR55 Is Highly Increased during the Progression of Alzheimer’s Disease in *App^NL-G-F^* Knock-In Mice

**DOI:** 10.3390/biology12060805

**Published:** 2023-05-31

**Authors:** Dina Medina-Vera, Hongjing Zhao, Erika Bereczki, Cristina Rosell-Valle, Makoto Shimozawa, Gefei Chen, Fernando Rodríguez de Fonseca, Per Nilsson, Simone Tambaro

**Affiliations:** 1Instituto de Investigación Biomédica de Málaga-IBIMA, Unidad de Gestión Clínica de Salud Mental, Hospital Regional Universitario de Málaga, 29010 Málaga, Spain; dina.medina@ibima.eu (D.M.-V.); cristina.rosell@ibima.eu (C.R.-V.); fernando.rodriguez@ibima.eu (F.R.d.F.); 2Facultad de Ciencias, Universidad de Málaga, 29010 Málaga, Spain; 3College of Wildlife and Protected Area, Northeast Forestry University, Harbin 150040, China; zhaohongjing@nefu.edu.cn; 4Department of Neurobiology, Care Sciences and Society, Division of Neurogeriatrics, Center for Alzheimer Research, Karolinska Institutet, 17164 Solna, Sweden; erika.bereczki@ki.se (E.B.); makoto.shimozawa@ki.se (M.S.); per.et.nilsson@ki.se (P.N.); 5Department of Biosciences and Nutrition, Karolinska Institutet, 14152 Huddinge, Sweden; gefei.chen@ki.se

**Keywords:** endocannabinoid system, cannabinoid receptors, CB2, GPR55, neuroinflammation, APP knock-in mice, Alzheimer’s disease

## Abstract

**Simple Summary:**

Alzheimer’s disease (AD) is a complex, multifactorial disease where numerous components, such as environment, lifestyle, comorbidities, and genetic predisposition, contribute to triggering the onset of the disease. Several neurobiological brain alterations have been reported during AD pathologies, including the endocannabinoid system (ECS) and associated lipid transmitter-based signaling systems. In this study, we have evaluated the expression levels of the cannabinoid receptors type 2 (CB2) and the novel cannabinoid/lysophospholipid G protein-coupled receptor 55 (GPR55) at different stages of AD. We deeply investigated CB2 and GPR55’s close proximity with Aβ-plaque deposits, as well as neuronal and glial cells, in the AD *App^NL-G-F^* knock-in mouse model. Additionally, we analyzed whether Aβ42 directly affects CB2 and GPR55 protein expression in neuronal and glial primary cell cultures. Our study shows that the ECS, specifically the CB2 and GPR55 receptors, are altered during AD pathology. Monitoring these receptors may provide new biomarkers for AD diagnosis. CB2 and GPR55 could be potential pharmacological targets for selective compounds to treat AD inflammation.

**Abstract:**

Background: The endocannabinoid system (ECS) and associated lipid transmitter-based signaling systems play an important role in modulating brain neuroinflammation. ECS is affected in neurodegenerative disorders, such as Alzheimer’s disease (AD). Here we have evaluated the non-psychotropic endocannabinoid receptor type 2 (CB2) and lysophosphatidylinositol G-protein-coupled receptor 55 (GPR55) localization and expression during Aβ-pathology progression. Methods: Hippocampal gene expression of CB2 and GPR55 was explored by qPCR analysis, and brain distribution was evaluated by immunofluorescence in the wild type (WT) and APP knock-in *App^NL-G-F^* AD mouse model. Furthermore, the effects of Aβ42 on CB2 and GPR55 expression were assessed in primary cell cultures. Results: CB2 and GPR55 mRNA levels were significantly upregulated in *App^NL-G-F^* mice at 6 and 12 months of age, compared to WT. CB2 was highly expressed in the microglia and astrocytes surrounding the Aβ plaques. Differently, GPR55 staining was mainly detected in neurons and microglia but not in astrocytes. In vitro, Aβ42 treatment enhanced CB2 receptor expression mainly in astrocytes and microglia cells, whereas GPR55 expression was enhanced primarily in neurons. Conclusions: These data show that Aβ pathology progression, particularly Aβ42, plays a crucial role in increasing the expression of CB2 and GPR55 receptors, supporting CB2 and GPR55 implications in AD.

## 1. Introduction

Neuroinflammation represents a prominent pathological component in many neurodegenerative diseases, such as Alzheimer’s disease (AD) [1,2]. The neuroinflammatory response in AD is primarily triggered by amyloid beta (Aβ) peptides in particular in their oligomeric form and as Aβ neuritic plaques, along with the intracellular accumulation of tau protein neurofibrillary tangles [1,3]. The amyloid cascade consists of the chronological onset of increased levels of amyloid and tau and evidence that Aβ overproduction is related to the onset of AD. According to this theory, the accumulation of Aβ leads to several concatenated events, culminating in neuronal damage, memory impairment, and neuronal death [4]. However, mounting evidence suggests that the amyloid cascade alone cannot account for much of the pathophysiology of AD, implying that other pathogenic mechanisms are involved [5]. With the finding of high levels of proinflammatory markers in patients with AD and AD risk genes linked to innate immune activities, inflammation has emerged to have a critical role in AD pathophysiology [6,7,8].

The endocannabinoid system (ECS) is widely regarded as a component of the central nervous system (CNS) endogenous neuroprotective processes and has been considered a therapeutic target for neuroinflammation [9,10]. In various pre-clinical models and human investigations, the endocannabinoid system, including cannabinoid receptor type 1 (CB1) and 2 (CB2), as well as the lysophosphatidylinositol G-protein coupled receptor 55 (GPR55), endogenous cannabinoids, and enzymes that catabolize these compounds, are involved in the development of neuroinflammatory illnesses, such as AD [11]. The CB2 and GPR55 are cannabinoid receptors present in both the glial cells and neurons [12,13]. Both of them are abundantly expressed not only in the CNS but also in the peripheral tissue [14,15]. During neuroinflammation and neurodegeneration, CB2 regulates microglial migration and penetration into brain regions [16], and CB2 activation may benefit AD by reducing local, microglia-mediated inflammation and improving Aβ elimination [17].

The GPR55 is an orphan G-protein-coupled receptor first described in 1999 [18]. Although GPR55 is considered a cannabinoid receptor because it displays an affinity for natural and synthetic cannabinoids, physiologically it is not an endocannabinoid receptor but a lysophosphatidylinositol receptor. Therefore, it can be activated by both cannabinoids and non-cannabinoid ligands, mainly lysophosphatidylinositol (LPI), leading to the hypothesis that it is a type-3 cannabinoid receptor [19]. The GPR55 signaling has been linked to both neurodegeneration and the modulation of specific cytokines and inflammation [20]. Most importantly, studies in mice lacking GPR55 have reported a reduction in inflammatory and neuropathic pain [21]. Furthermore, CB1 and CB2 have been found in senile plaques in AD patients which are linked to microglial activation [22]. While CB1 expression is significantly reduced in AD brains [22], CB2 levels are shown to increase in AD patients and correlate with the brain levels of Aβ42 and senile plaque score [23]. The pronounced ECS alterations, as detected in AD patients, were also observed in several AD mouse models. In this context, identifying when the endocannabinoid (ECB) signaling system is altered in AD could potentially provide early biomarkers of brain dysfunction suitable for aiding the clinical diagnosis of AD. Regarding LPI and phosphatidylinositol (its main precursor), a recent publication demonstrated a clear correlation between changes in these lipids and Aβ42 load in humans [24]. On the other hand, pharmacological modulation of these lipid transmitters by using selective drugs interacting with CB2 and GPR55 receptors might prevent or delay neurodegeneration and the subsequent onset of clinical alterations and dementia.

In a previous study, we demonstrated an imbalance of the ECS in the transgenic mouse model 5xFAD (FAD: familial Alzheimer’s disease) [25,26]. Specifically, we described that the increase in CB2 and GPR55 receptors in the hippocampus of homozygous 5xFAD mice was associated with emotional and memory performance. Additionally, the increased expression of CB2 and GPR55 receptors positively correlated with a rise in neuroinflammatory markers [27]. However, although the 5xFAD animals display strong neuroinflammation (astrogliosis and microgliosis) in response to amyloid deposition [26,28], these mice are overexpressing the amyloid precursor protein (APP). This results in an overproduction of other APP fragments in addition to Aβ peptides [29]. For this reason, we have here, for the first time, studied the ECS in a mouse model with a typical Aβ pathology and neuroinflammation, but without APP overexpression. Hence, we used a recently developed APP knock-in AD mouse model: the *App^NL-G-F^*. Since the natural APP expression pattern is physiological in the *App^NL-G-F^* mice, this mouse model is more congruent with AD’s human pathophysiology and clinical elements. It also reduces the risk of artifacts by keeping physiological APP expression at endogenous levels [29]. As a result, we can mimic most of the Aβ-neuroinflammation-related diseases seen in patients using our mouse model.

In this study, we have investigated the expression levels of the cannabinoids/Lysophospholipids receptors, CB2 and GPR55, at different stages of AD-like pathologies. CB2 and GPR55 overlapping staining with Aβ-plaque deposits, neuronal and glial cells were thoroughly investigated in 12-month-of-age *App^NL-G-F^* mice. Furthermore, we have analyzed whether Aβ42 treatment of neuronal and glial primary cell cultures would directly affect the CB2 and GPR55 protein expression.

## 2. Materials and Methods

### 2.1. Animals and Ethics Statement

We used wild-type (WT), and APP knock-in *App^NL-G-F^* mice harboring the Swedish “*NL*”, the Beyreuther/Iberian “*F*” and the Arctic “*G*” mutation in the App gene. The Swedish and the Beyreuther mutations raise the ratio of Aβ42/Aβ40 and result in high levels of Aβ -toxicity. The Arctic promotes Aβ oligomerization [30]. All experiments included both females and males at 3, 6, and 12 months of age, and were performed in compliance with the ARRIVE guidelines [31] and concordance with the European Communities Council Directives 2010/63/EU, Regulation (EC) No. 86/609/ECC (24 November 1986). A total of 3 mice across the genotypes and ages were used in this study reflecting our efforts to minimize animal suffering and reduce the number of animals used. The animal experiments were performed under the ethical permit 15758-2019 obtained from the Stockholm ethical board.

### 2.2. RNA Isolation, and qPCR

The RNA was extracted from dissected hippocampal tissue preserved in RNAlater (AM7020, Thermo Scientific, Waltham, MA, USA) using RNeasy Lipid Tissue Mini Kit (74804, Qiagen, Germantown, MA, USA) according to the manufacturer’s instructions. The RNA quality (RNA integrity number) and quantity were measured by a Bioanalyzer 2100 (Agilent, Santa Clara, CA, USA) using the Agilent RNA 6000 Nano Kit (cat number 5067-1511). 200 ng of RNA for each sample were reverse transcripted to cDNA. The RNA was mixed with 20μL of master mix and amplified by using a cycler (S1000 Thermal cycler Bio-Rad, Hercules, CA, USA). For the RT-qPCR a total of 10 μL reactions were run in duplicates using TaqMan Fast advanced Mastermix (Thermo Fisher, Waltham, MA, USA) and StepOne Plus real-time PCR Detection System (Applied Biosystem, Waltham, MA, USA). The primers (Thermo Fisher, Waltham, MA, USA) used in this study were: Cnr2 (Mm02620087_s1), Gpr55 (Mm02621622_s1), and Tubb3 (Mm00727586_s1). The expression levels of the genes of interest were normalized against β-Tubulin. Relative quantification (RQ) for mRNA was calculated using the ∆∆ cycle threshold (∆∆CT) method, with fold changes using the formula, ΔΔCt = ΔCtT2 − ΔCtT1.

### 2.3. Histological and Immunofluorescence Analysis

Mice (WT and *App^NL-G-F^*) were anesthetized with isoflurane and intracardially perfused with phosphate-buffered saline (PBS). After the brains were quickly removed and put in paraformaldehyde (PFA) 4%. Coronal sections at 5 μm thickness were obtained from paraffin-embedded hemibrains using a microtome. Sections were de-paraffinized by washing in xylene and re-hydrated in decreasing concentrations of ethanol (from 99% to 70%). For antigen retrieval, sections were pressure boiled in an autoclave immersed in citrate buffer at 110 °C for 5 min. After a 30 min incubation with TNB blocking buffer, (TSA Blocking Reagent Cat#FP1020, Akoya Biosciences, Marlborough, MA, USA) sections were incubated at 4 °C O/N with primary antibodies diluted in TNB blocking buffer at the following concentration: polyclonal rabbit anti-G protein-coupled receptor 55 (GPR55) 1:500 (r) and polyclonal rabbit anti-cannabinoid type 2 receptor (CB2) 1:250 (Cat# ab3561, Abcam, Cambridge, UK). Negative control slides lacking the first antibody were used for every staining. After thorough washing in phosphate-buffered saline with 0.05%Tween 20 (PBST), sections were incubated with biotinylated goat anti-rabbit secondary antibody (Cat#BA-1000, Vector Laboratories Inc., Newark, CA, USA) diluted 1:200 in TNB blocking buffer at RT for 2 h and after a washing step, sections were incubated with Streptavidin–HRP reagent (TSA Fluorescein System Kit, Cat#NEL701A001KT, Akoya Biosciences, Marlborough, MA, USA) diluted 1:100 in TNB blocking buffer for 30 min. Thereafter, slides were washed in PBST and incubated with fluorophore Tyramide (TSA Fluorescein System Kit, Cat#NEL701A001KT, Akoya Biosciences, Marlborough, MA, USA)) diluted 1:50 in Amplification Reagent (TSA Fluorescein System Kit, Cat#NEL701A001KT, Akoya Biosciences, Marlborough, MA, USA)) for 10 min. For Aβ plaque detection, slides were further incubated in 1-fluoro-2,5-bis(3-carboxy-4-hydroxystyryl) benzene (FSB) (Cat#344101-5MG, Millipore, Burlington, MA, USA) diluted 1:1000 in PBST at RT for 30 min. Lastly, slides were washed in PBST and mounted with Vectashield HardSet Antifade Mounting Medium with DAPI (Cat#H-1500, Vector Laboratories Inc., Newark, CA, USA) or PermaFluor water-soluble mounting media (Cat#TA-030-FM, Thermo scientific, Waltham, MA, USA ) if FSB staining was completed.

For double staining after incubation with fluorophore Tyramide for 10 min and washing the slides with PBST, a second primary antibody for microglia (rabbit anti-Iba1, Cat#CAK1997, Fujifilm Wako, Osaka, Japan) or astrocytes (rabbit anti-GFAP, Cat#Z0334, DAKO, Santa Clara, CA, USA), diluted 1:100 in TNB blocking buffer was added at +4 °C O/N. After thorough washing, the sections were incubated with Alexa Fluor 546 conjugated goat anti-rabbit secondary antibody (Cat#A-11035, Thermo Fisher, Waltham, MA, USA) diluted at 1:500 in TNB blocking buffer at RT for 2 h followed by washing and mounting steps previously described.

### 2.4. Primary Cell Cultures

Mixed cortex and hippocampi tissues from mouse embryos (E17-E18) were dissected under the microscope in Hanks’ Balanced Salt Solution (HBSS, Gibco, Waltham, MA, USA) for culturing primary neurons. After transfer into a new tube, the digestion of cortex and hippocampi tissues was performed using accutase (Cat #A1110501, Gibco, Waltham, MA, USA) at 37 °C for 10 min and terminated using 10 mL Neurobasal Media (Cat #A1371201, Gibco, Waltham, MA, USA) containing GLUTAMAX 1% (Cat #35050061, Gibco, Waltham, MA, USA) and B27 2% (Cat #17504044, Gibco, Waltham, MA, USA). After centrifugation at 1000 rpm for 3 min, the supernatant was carefully removed, leaving cells suspended with 5 mL Neurobasal Media and filtered into the new tube through a cell strainer (70 μm). Finally, neurons were added to a 24-well poly-D-lysine-containing plate after counting.

For culturing the astroglia-microglia primary cell, mouse embryo brains dissections were dissolved with 10 mL media (DMEM/F12 (Cat #11320033, Gibco, Waltham, MA, USA) + 10% FSB + N2 (1:100) (Cat #17502048, Gibco, Waltham, MA, USA)). After filtering through a cell strainer, the cells were then plated in a petri dish and incubated at 37 °C. After 2 weeks, astrocyte and microglia cells were isolated as previously described by Saura and colleagues in 2003 [32]. Briefly, the co-cultures astroglia-microglia were washed with DPBS and treated with 2 mL Trypsin 0.08% EDTA with DMEM-F12 until the formation of a detached cell layer (Astrocytes layer). The detached cells were collected and diluted with 5 mL of DMEM-F12 + 10% FSB, centrifuged at 900× *g* rpm for 5 min, counted, and replated in 24-well plates for the immunocytochemistry studies. The remaining microglial cells in the petri dishes were detached by using 2 mL of Trypsin 0.25%. The solution containing the detached cells was collected and diluted with 5 mL of DMEM-F12 + 10% FSB and centrifuged at 1800 rpm for 5 min. Finally, microglia were re-suspended with DMEM-F12 and added to the 24-well plate after counting.

### 2.5. Aβ42 Production

Recombinant Aβ42 (Met-Aβ residues 1–42) peptide was prepared according to previously published protocols (45). Briefly, the Aβ42 peptide was produced in BL21(DE3) pLysS *Escherichia coli* and purified using DEAE-Sepharose (GE Healthcare, Chicago, IL, USA). The eluted fraction was filtered through a 30,000 Da Vivaspin concentrator (4000× *g*, GE Healthcare, Chicago, IL, USA) at 4 °C to remove large Aβ42 aggregates. The resulting filtrate was concentrated to ~50 μM at 4 °C using a 5000 Da Vivaspin concentrator (4000× *g*, GE Healthcare, Chicago, IL, USA). The concentration of Aβ42 peptide was determined using an extinction coefficient of 1400 M^−1^cm^−1^. Finally, the Aβ42 peptides were aliquoted into low-bind Eppendorf tubes (Axygene, Union City, CA, USA) and stored at −80 °C for further use.

### 2.6. Cell Treatment

Before incubation, Aβ42 was filtered using a 0.22 μm (Millipore, Burlington, MA, USA) filter in order to remove any unwanted bacterial protein. Before the Aβ42 treatment, the cell media was removed from each plate and replaced with DMEM/F12 with 1% of FSB. Cells were then treated with Aβ42 1 μM. All treatments were carried out in triplicates and incubated for 24 h. After incubation cells were collected for immunocytochemistry analysis.

#### Immunocytochemistry

Primary neurons, microglial, and astrocyte cells were seeded on glass coverslips in 24-well plates. After two weeks, cells were treated with Aβ42 for 24 h. After that, cells were fixed in 4% paraformaldehyde for 10 min and washed three times with PBS before permeabilization with PBS containing 0.1% Triton X-100 (10 min incubation). Next, washed three times with PBS. Cells were blocked for 1 h with PBS containing 3% bovine serum albumin (BSA). Cells were labeled with a rabbit anti-GPR55 antibody (diluted 1/200 in 3% BSA/PBS; Cat#10224, Cayman, Ann Arbor, MI, USA) and a rabbit anti-CB2 antibody (diluted 1/200 in 3% BSA/PBS; Cat# ab3561, Abcam), and incubated at 4 °C O/N. Negative control slides lacking the first antibody were used for every staining. To evaluate the purity of the astroglia cells, we used anti-GFAP (diluted 1:500 in 3% BSA/PBS; Agilent Technologies, Santa Clara, CA, USA), and for microglia cell culture, anti-Iba1 (diluted 1:250 in 3% BSA/PBS; Fujifilm Wako, Osaka, Japan). After thorough washing in PBS three times, the cells were incubated with a secondary antibody anti-rabbit AlexaFluor (Invitrogen, Waltham, MA, USA) diluted 1:500 in 3% BSA/PBS at RT for 2 h. The cells were, after a washing step, incubated with Hoechst staining solution (diluted 1:500 in PBS) for 15 min at RT. After, the samples were washed three times and mounted with PermaFluor mounting media (Epredia-Thermo Scientific, Waltham, MA, USA) for visualization.

### 2.7. Microscopy and Fluorescence Intensity Quantification

Images were acquired with a digital camera (Nikon D5-Qi2) connected to a Nikon fluorescence microscope (Nikon Eclipse E800) with a Plan-Apochromate 2×, 4×, 10×, and 20× objectives. NIS-Elements D software, version 4.30.00, was used for image processing. We used ImageJ software [33] for fluorescence intensity quantification. All pictures were taken with the same excitation light intensity, exposure time, and analog gain. The images were then binarized to 8-bit black, and a fixed intensity threshold was applied for each immunostaining. We outlined the region of interest with the ROI tool and set Area, Integrated Density, and Mean Grey Value as the desired parameters to analyze. Four mice per group and two sections per mouse were used. For the primary cell, experiments were performed at least in triplicates, and from each slide, at least 2 images were used for quantification. For better image quality (not included in the quantification), selected images were taken also with Zeiss LSM800 using ZEN software, when oil immersion with a 63× objective was used to improve imaging.

### 2.8. Statistical Analyses

Analyses were performed using the GraphPad Prism (version 9). The Shapiro–Wilk test was used to assess the normal distribution of data. For qPCR experiments, two-way analysis of variance (ANOVA) was assessed for comparisons among the different animal groups, followed by Tukey’s multiple comparisons test. The analysis of two single groups was performed using Student’s unpaired *t*-test. All image quantification data are presented as the mean ± standard error of the mean (SEM) or as stated in the figure legends, and *p*-values less than 0.05 were considered statistically significant.

## 3. Results

### 3.1. CB2 Expression Significantly Increases at 6, and 12 Months of Age in App^NL-G-F^ Mice

To evaluate the expression levels of the endocannabinoid receptors CB2 (*Cnr2*) and GPR55 (*Gpr55*) during the course of AD pathology, we have evaluated and compared their hippocampal gene expression in wild-type (WT) and the *App^NL-G-F^* mice at 3, 6, and 12 months of age. The qPCR results obtained reveal that CB2 gene expression (*Cnr2*) is upregulated in 6 and 12 months old *App^NL-G-F^* when compared with the healthy control WT mice (Figure 1a) (*p*-value < 0.05). At 2 months, when the Aβ pathology is still very low in the *App^NL-G-F^* AD mouse model, CB2 gene expression levels did not significantly differ between WT and *App^NL-G-F^* mice. These data strongly support a direct correlation between CB2 gene expressions and the severity of AD pathology. Therefore, Aβ pathology harbored by *App^NL-G-F^* mice might be a critical factor in promoting gene expression of CB2 and GPR55 receptors at an advanced age.

Furthermore, the expression and distribution of the CB2 receptor in *App^NL-G-F^* mice at the age of 12 months, when the Aβ deposition is nearly saturated with a concomitant developing high microgliosis and astrocytosis, were further characterized. Immunostaining for CB2 receptors confirmed a higher expression of CB2 in the hippocampus of the *App^NL-G-F^* mice, specifically in the *Cornu Ammonis* 3 and 1 area (CA3 and CA1) compared to WT mice (*p* < 0.01; Figure 1b). Most importantly, positive CB2 immunostaining was widely distributed in the hippocampus of the *App^NL-G-F^* mice, whereas in the WT group CB2 positive staining was detectable only in the dentate gyrus (DG). Overall, the distribution and pattern of the CB2 receptors immunostaining in *App^NL-G-F^* compared to the WT mice were substantially different; in control mice (WT) positive CB2 staining was in hilus of DG, whereas in *App^NL-G-F^* mice, CB2 staining was observed also in granular staining accumulations (Figure 1b, white arrows).

To evaluate the cell specificity of the CB2 staining surrounding the Aβ plaques, double immunofluorescence staining for CB2 and selective markers for astrocytes (GFAP), microglial (Iba1) respectively, and amyloid plaque (FSB) markers was performed. High CB2-positive staining was found in and around astrocytes (Figure 2a) in Aβ-plaques deposits, where a prominent accumulation of CB2-expressing activated astrocytes surrounding the plaques was observed (Figure 2b).

The microglia co-staining showed, in *App^NL-G-F^* mice, a high immunoreactivity of CB2 in microglia surrounding the FSB-positive Aβ-plaques (Figure 2c,d). Therefore, CB2 immunostaining revealed positive CB2 staining within activated microglia in the proximity of Aβ plaques. Hence, we conclude that in microglia and astrocyte surrounding the Aβ plaques CB2 receptors are highly expressed.

### 3.2. Increased GPR55 mRNA Levels and Immunoreactivity in App^NL-G-F^ Mice

The GPR55 gene expression levels were evaluated by qPCR analysis. The data showed higher GPR55 mRNA expression levels in *App^NL-G-F^* mice at 6 and 12 months compared to WT mice (Two-way ANOVA: *p* < 0.01; Figure 3a).

Brain localization and expression of GPR55 in *App^NL-G-F^* mice at the age of 12 months were further evaluated by immunohistochemistry analysis. Immunostaining of GPR55 receptors revealed an increased expression of GPR55 in the hippocampus of *App^NL-G-F^* mice, specifically in the CA3, DG, and CA1 (*p* < 0.001), as compared to healthy control mice (Figure 3b). In the CA3 region, GPR55 immunoreactivity was significantly increased in the AD mice’s pyramidal cell layer (PCL). In the dentate gyrus, a higher immunoreactivity of GPR55 was detected in the granular cell layer (GCL) and the hilus (H). In the hippocampal CA1 region of the *App^NL-G-F^*, GPR55 was highly detected in the stratum pyramidal (SP) and the stratum lacunosum (SL). The GPR55 immunoreactivity was not only hippocampal specific but it was also observed in the prefrontal cortex (PFC) and basolateral amygdala (BLA) areas. While no difference was observed between the WT mice and the *App^NL-G-F^* in the PFC region (ns; Appendix A), a significantly higher expression of GPR55 was detected in the basolateral amygdala (BLA) of the *App^NL-G-F^* compared with the WT mice group (*p* < 0.001; Figure 3c).

Differently to CB2, no immune overlapping staining between GPR55 and the astrocytes marker GFAP was detected (Figure 4a), instead, GPR55 co-localized with the microglia marker Iba1 (Figure 4b). To summarize, GPR55 positive staining was detected in neuronal cells and microglia, but not in astrocytes. In particular, an accumulation of GPR55 staining was observed in the microglia surrounding the FSB-positive Aβ plaques (Figure 4c).

### 3.3. Aβ42 Treatment Induces CB2 and GPR55 Overexpression in WT Mouse Primary Neuronal Cell Culture, and CB2 in Primary Glial Cell Culture but Not for GPR55

To further investigate the effect of Aβ42 on CB2 and GPR55 expression, neurons, microglia, and astrocytes primary cultures derived from WT mice were exposed to Aβ42. The purity of astroglial and microglial cell cultures was validated by using the specific cell markers GFAP and Iba1, respectively (Appendix A)**.** Aβ42 treatment affected the CB2 level in WT neurons (*p* < 0.05; Figure 5a) and the treatment also induced a significant increase in expression in both astrocytes (*p* < 0.001; Figure 5b) and microglia cells (*p* < 0.001; Figure 5c).

For the GPR55 protein, we found significant changes in Aβ42 treated neurons (*p* < 0.001; Figure 6a). However, neither astrocytes nor microglia showed any alterations after Aβ42 exposure (ns; Figure 6a,b). It may indicate that Aβ42 affects the expression of GPR55 mainly in neurons, whereas Aβ42 affects CB2 expression in both neurons and glial cells.

### 3.4. Aβ42 Exposure Induced Upregulation of CB2 and GPR55 in App^NL-G-F^ Mouse Primary Neuronal Cell Culture and in Primary Glial Cell Culture for CB2 But Not for GPR55

Having found that Aβ exposure increased expression of CB2 and GPR55 in cells derived from WT mouse brains, we next asked the question of whether cells derived from the brains of App^NL-G-F^ mice, which have been exposed to Aβ pathology, exhibited altered sensitivity to Aβ. As observed in neurons, astrocytes, and microglia from WT mouse brains, Aβ42 treatment induced an up-regulation of CB2 in neurons (*p* < 0.001; Figure 7a), in astrocytes (*p* < 0.001; Figure 7b), and in microglial (*p* < 0.05; Figure 7c) cells after 24 h treatment.

Aβ42 treatment of *App^NL-G-F^* primary cell cultures also induced a significant upregulation of the GPR55 protein expression in neurons derived from *App^NL-G-F^* mice (Figure 8, negative controls in Appendix A). Notably, in contrast to the effects seen in microglia derived from WT mouse brain, Aβ42 induced an increase in GPR55 levels in microglia (*p* < 0.05; Figure 8a and Figure 8c, respectively), whereas no changes were detected in *App^NL-G-F^* astrocytes after Aβ42 treatment (ns; Figure 8b).

In summary, in Aβ42-treated *App^NL-G-F^* cell cultures, CB2 protein expression significantly increased in both primary neurons and glial cells, as observed in primary WT cell cultures. On the other hand, GPR55 protein expression after Aβ42 treatment was differently affected comparing primary cell cultures from the brains of WT and *App^NL-G-F^* mice; While GPR55 was considerably elevated in both WT and *App^NL-G-F^* generated neurons, GPR55 was particularly raised in microglia cells derived from *App^NL-G-F^* mice (Appendix A) but not WT animals. These results show that both receptors undergo changes upon Aβ42 exposure and are implicated in AD.

CB2 and GPR55 staining intensity was further evaluated by comparing the effect of the genotype and treatment. Two-way ANOVA analysis (Appendix A) revealed that genotype is influencing (WT vs. *App^NL-G-F^* mouse primary neuronal cell culture), the amount of CB2 fluorescence intensity in neurons. CB2 intensity is significantly higher in neurons derived from *App^NL-G-F^* mice as compared to those from WT mice (Two-way ANOVA and Tuckey’s multiple comparison test: interaction: *p* = 0.0033, genotype factor: *p* < 0.0001). Moreover, when the Aβ42 treatment is added, the CB2 expression level in neurons is even higher (Two-way ANOVA and Tuckey’s multiple comparison test: treatment factor: *p* < 0.0001). However, we did not see the same effect for the GPR55 in neurons (Two-way ANOVA and Tuckey’s multiple comparison test: interaction: non-significant, genotype factor: *p* < 0.0001), where the GPR55 receptor basal levels are the same. Nevertheless, after the Aβ42 treatment, the GPR55 levels in the *App^NL-G-F^* neuronal cells are higher than those in Aβ42-treated neurons from WT mice (Two-way ANOVA and Tuckey’s multiple comparison test: treatment factor: *p* = 0.011) which was similarly observed in WT-treated microglia (Two-way ANOVA and Tuckey’s multiple comparison test: interaction: non-significant, genotype factor: *p* = 0.0217, treatment factor: *p* < 0.01).

## 4. Discussion

AD is a complex multifactorial disease to which numerous components such as environment, lifestyle, comorbidities, and genetic predisposition contribute to triggering the onset of the disease [34]. Several neurobiological brain alterations have been reported during the course of AD pathology, including the ECS [35] and LPI-related lipids. To date, the pathophysiological role of the ECS/LPI in AD is still unknown, as it is unclear at which stage of the disease starts its alteration. Moreover, these alterations collectively align with the described disrupted lipid homeostasis in the brain that might trigger/promote/accelerate AD pathology (the lipid invasion model [36]).

Most of the research on these lipids has focused on the ECS receptor CB1 [37,38,39]. CB1 is considered one of the most abundant G-protein-coupled receptors expressed in mammalian neurons in the brain, and its activation is linked to the psychotropic effect of marijuana, but CB1-activation is also linked to neuroprotection and neurogenesis [40]. In AD post-mortem tissues CB1 activity and expression were found significantly decreased in several brain regions [41,42]. On the other side, the expression and role in AD pathophysiology of the non-psychotropic receptors CB2 and, even further for GPR55 remain to be elucidated. For that purpose, the present study explored the expression and brain localization of these two lipid membrane receptors in physiological and pathophysiological conditions by using WT and the AD *App^NL-G-F^* knock-in mice focusing on the hippocampal region, based on its early implication in AD [43].

Hippocampal gene expression analysis showed a significant upregulation of both CB2 (Cnr2) and GPR55 (Gpr55) mRNA in the AD *App^NL-G-F^* mice at 6 and 12 months compared with same-age healthy control WT mice. Therefore, considering the Aβ pathology that *App^NL-G-F^* mice harbor, the increased CB2 and GPR55 gene expression correlate positively with high levels of Aβ load. *App^NL-G-F^* mice develop an aggressive Aβ amyloidosis in an age-dependent manner starting at 2 months and almost saturated by 7 months [29]. Furthermore, in line with the gene expression data, we have observed an elevated hippocampal CB2 and GPR55 immunoreactivity in *App^NL-G-F^* versus the WT mice group at 12 months of age. This is an age where *App^NL-G-F^* mice have a strong memory and cognitive dysfunction, and most importantly, they display extensive microgliosis and astrocytosis, especially in the vicinity of plaques [44]. The CB2 data align with previous studies where increased protein levels or immunoreactivity were reported in other AD patients [23,45].

This significant hippocampal increase in GPR55 levels in *App^NL-G-F^* mice found in this study confirms our previous findings, where for the first time, we have reported a strong hippocampal upregulation of GPR55 in another AD mouse model (5xFAD) [27]. These data together strongly support an alteration of GPR55 levels in AD pathology, and most importantly, the expression of GPR55 correlates with the progression and severity of Aβ pathology. This finding matches the recent description of LPI alterations in postmortem human brains from AD patients, where they correlate with Aβ42 load [24]. Furthermore, it has been reported that hippocampal GPR55 stimulation with lysophosphatidylinositol improved synaptic plasticity, enhancing CA1 LTP [46]. Recently, by using a GPR55 agonist, O-1602, Xiang and colleagues showed attenuation in cognitive impairment, neurotoxicity neuroinflammation, and synaptic dysfunction in a mouse model of AD induced by Aβ42 and streptozotocin [47,48]. These findings support that a GPR55 pharmacological modulation might be a potential strategy for the therapy of neurodegenerative illnesses such as AD [48].

Another significant finding of this study is the higher GPR55 expression in the basolateral amygdala of the *App^NL-G-F^* mice compared with the WT mice group, suggesting a potential involvement in anxiety and fear. Anxiety is present in about 40% of AD patients and often occurs early in the course of AD [49]. In previous works, GPR55 activation has been associated with an amelioration of anxiety-like symptoms in mice [50], similarly, a GPR55 modulation of anxiety-like behaviors was also observed in rats [51]. In this regard, a recent report on AD patients showed that cannabidiol (CBD) may be an effective and safe choice for managing the behavioral and psychological symptoms of dementia [52]. Future clinical trials are needed to reassure these findings. Taking all this information together, we speculate that a selective GPR55-modulation can be beneficial to treat neuropsychiatric symptoms associated with AD. Interestingly, GPR55 has been described as a relevant modulator of insulin secretion and glucose tolerance because of its functional expression in pancreatic islets. Because of the proposed contribution of insulin resistance to AD progression, this further support the need of exploring the GPR55 contribution to AD [53].

Our data confirm the presence of a significant astrocytosis and microgliosis in *App^NL-G-F^* knock-in mice hippocampus at 12 months of age, which is associated with a high CB2 receptor expression in astrocytes and microglia surrounding the Aβ plaques. Instead, GPR55 was detected in microglia but not in astrocytes. Similar to the expression pattern of CB2, these activated microglia expressing GPR55 receptors were found around amyloid plaques. These findings demonstrate that both receptors undergo changes upon Aβ oligomerization and aggregation, and therefore, both are linked to AD pathology.

Neuroinflammation is one of the most important aspects of AD pathogenesis, and several pro-inflammatory and neurotoxic substances are released from activated microglia as a consequence of cytotoxic stimulatory factors such as Aβ fibrils and Aβ plaque deposition [54]. This results in increased neuronal death and degeneration, which secretes neurotoxic substances and enhances the vicious AD cycle [55]. Studies have demonstrated that CB2 activation can reduce neuroinflammation, in part by reverting the microglia-activated state [56,57]. These findings suggest that CB2 plays a critical role in restoring microglial homeostasis. Therefore, targeting the CB2 receptors could be a potential therapeutic approach to reduce AD neuroinflammatory processes. Chronic administration of selective CB2 agonists resulted in reduced TNF-α levels and lower Aβ-plaque load with improvement in cognitive performance [58,59,60]. Furthermore, a CB2 activation is also associated with a reduction in neuroinflammation and clearance of Aβ-plaques as well as a restoration of cognition, synaptic plasticity, and memory, reduced expression of specific microglia markers, and decreased secretion of pro-inflammatory cytokines [61]. On the other hand, GPR55 activation has been reported to decrease the release of interleukins. GPR55 antagonists effectively block microglial activation, similarly, in GPR55 −/− knockout mice a reduction in the release of the proinflammatory cytokines has been observed [62,63,64]. This role of CB2 and GPR55 in modulating neuroinflammation and the high expression of both CB2 and GPR55 in microglia cells during AD pathology support the importance of these two receptors as potential therapeutic targets and support the need for more clinical studies. Furthermore, treating neuronal and glial cells derived from brains of WT and *App^NL-G-F^* mice with Aβ42, the most neurotoxic Aβ variant, induced a significant upregulation of CB2 expression in astrocytes and microglia cells but was also in neurons. GPR55 expression also significantly increases in Aβ42-exposed neurons from WT and *App^NL-G-F^* mice. Interestingly, GPR55 expression was specifically induced by Aβ42 in microglia from *App^NL-G-F^* mice but not from WT mice, indicating that the microglia in the *App^NL-G-F^* mice have been sensitized by the AD pathologies present in the brain. These data together support that Aβ42 directly modulates both CB2 and GPR55 expression. The data also showed a substantial increase in CB2 neuronal expression in pathological conditions.

Noteworthy, although CB2 and GPR55 are G protein-coupled receptors known to be primarily expressed on the cell membrane, intense nuclear staining was observed in both neurons and glial primary cell cultures. To date, the presence of these two receptors in the cell nucleus has not been established; however, in line with our results, some other studies have reported the presence of intracellular CB2 receptors, such as in endoplasmic reticulum and Golgi apparatus [65,66], and even the presence of CB2-positive staining in cell nuclei [67]. Similarly, there have been studies investigating the subcellular localization of the GPR55 receptor where it has been reported the expression of GPR55 intracellularly and in the nucleus [13,19,68]. Undoubtedly, further studies are needed to definitively validate the presence of these receptors in the cell nucleus.

While our results show alterations of cannabinoid receptors shown in the *App^NL-G-F^* knock-in mice, a limitation of using an animal model for such a complicated disease as AD must be considered when interpreting the data. However, the *App^NL-G-F^* mice exhibit neuroinflammation, synaptic changes, and progressive cognitive decline in addition to the Aβ-induced pathology [29,44,69], enabling us to reproduce most of the Aβ-related pathology seen in patients. Another aspect that strengthens the results of this study is related to the mouse model used. Most of the previous preclinical studies on the characterization of the ECB in AD have been conducted in AD transgenic mice overexpressing unphysiological amounts of human APP and displaying artefactual symptoms due to the overexpression of APP levels [29,70]. Therefore, recapitulating the CB2 and GPR55 alterations in a model with physiological APP expression is contributing to the understanding of the ECB system in AD. These mouse models, however, still have drawbacks because they show limited tau pathology associated with AD including the formation of neurofibrillary tangles [70]. Therefore, it would be interesting to investigate in the future, an animal model that also exhibits tau pathology especially since, animals lacking CB2 were shown to suffer mitochondrial malfunction, hippocampus-dependent memory impairment, and tau neuropathology [71].

## 5. Conclusions

With these findings, we have shown that the ECS, with the key players CB2 and GPR55 receptors, is altered in AD upon the development of the Aβ pathology. The monitoring of these receptors may be new biomarkers for AD early stages suitable for clinical diagnosis. Furthermore, CB2 and GPR55 can be potential pharmacological targets of selective compounds that modulate the signalling mechanisms including neuroinflammation.

However, further studies are necessary to understand the mechanisms behind the Aβ-induced upregulation of CB2 and CPR55. For a better characterization of the therapeutic importance of targeting CB2 and GPR55 in AD, treatment with selective compounds for those receptors will be necessary, alone or in combination.

## Figures and Tables

**Figure 1 biology-12-00805-f001:**
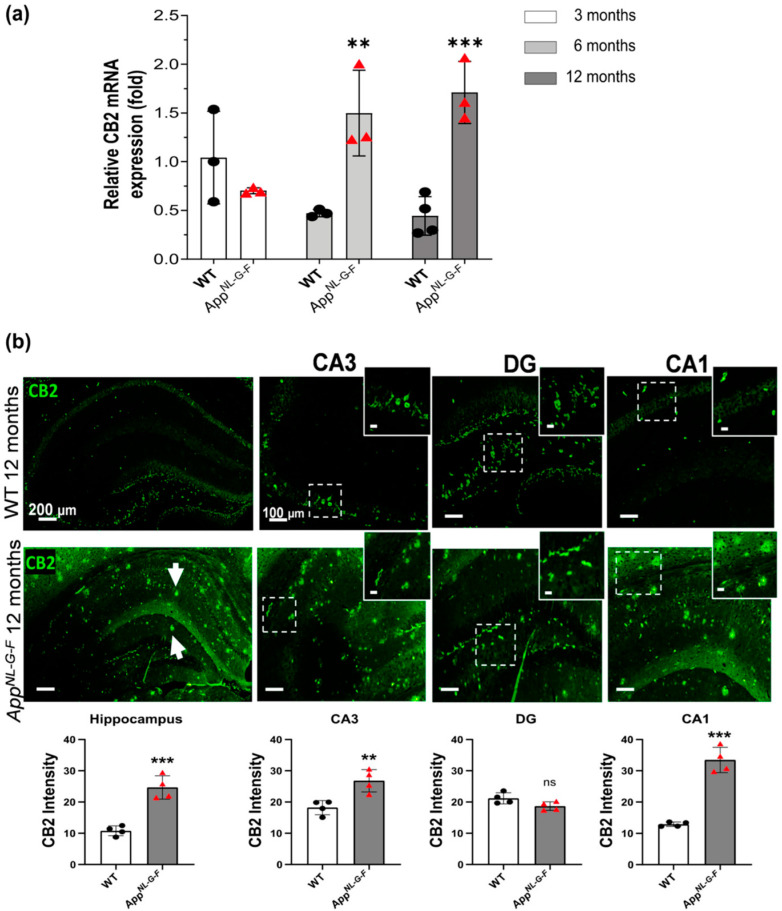
CB2 mRNA expression and immunoreactivity analysis in *App^NL-G-F^* mice and healthy control WT mice. (**a**) The histograms show the relative CB2 mRNA expression (fold) in WT and *App^NL-G-F^* mice at 3, 6 and 12 months old (*n* = 3 to 4). Data represent mean ± S.E.M. One-way ANOVA was used, ** denotes *p* < 0.01 and *** denotes *p* < 0.001. (**b**) Hippocampal CB2 immunoreactivity in 12-month-old *App^NL-G-F^* mice. Immunofluorescence images show the widespread distribution of CB2 receptors (green) in the hippocampus. The immunoreactivity was seen more specifically in the dentate gyrus (DG) of the WT, and in the CA3 and CA1 in *App^NL-G-F^* mice. Arrows point specific CB2 high accumulation areas in *App ^NL-G-F^* mice hippocampus. Scale bars represent 200 μm. Data represent mean ± S.E.M. (*n* = 4 mice per group and 2 sections per mouse were used for quantification). *t*-test was used, ns, not significant; ** denotes *p* < 0.01 and *** denotes *p* < 0.001.

**Figure 2 biology-12-00805-f002:**
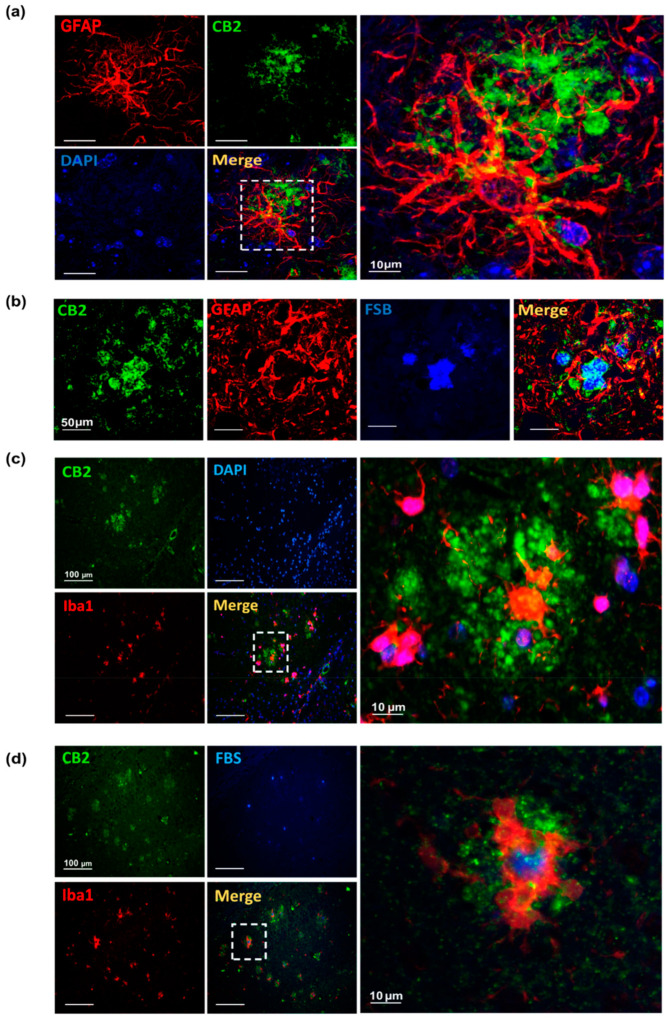
Astrocytes and microglia CB2 immunoreactivity in 12-month-old *App^NL-G-F^* mice. (**a**) Immunofluorescence images show astrocytes stained for GFAP (red), CB2 receptors (green) and the cell nuclei marker DAPI (blue) showing close proximity of GFAP and CB2 staining in the hippocampus. Scale bars represent 50 and 10 μm. (**b**) Representative images showing overlapping staining of CB2 (green) and amyloid plaques (FSB, blue) were observed. Also, astrocytes (GFAP, red) were found around these plaques. Scale bars represent 50 μm. (**c**) Representative immunofluorescence images immunostaining revealing overlapping staining between microglia marker (Iba1, red) and CB2 receptors (green), and (**d**) microglia marker (Iba1, red) and CB2 receptors (green) within the amyloid plaques (FSB, blue). Scale bars represent 50 and 10 μm. Abbreviations: GFAP: Glial fibrillary acidic protein, FSB: 1-Fluoro-2,5-bis(3-carboxy-4-hydroxystyryl) benzene, DAPI: (4′,6-diamidino-2-phenylindole).

**Figure 3 biology-12-00805-f003:**
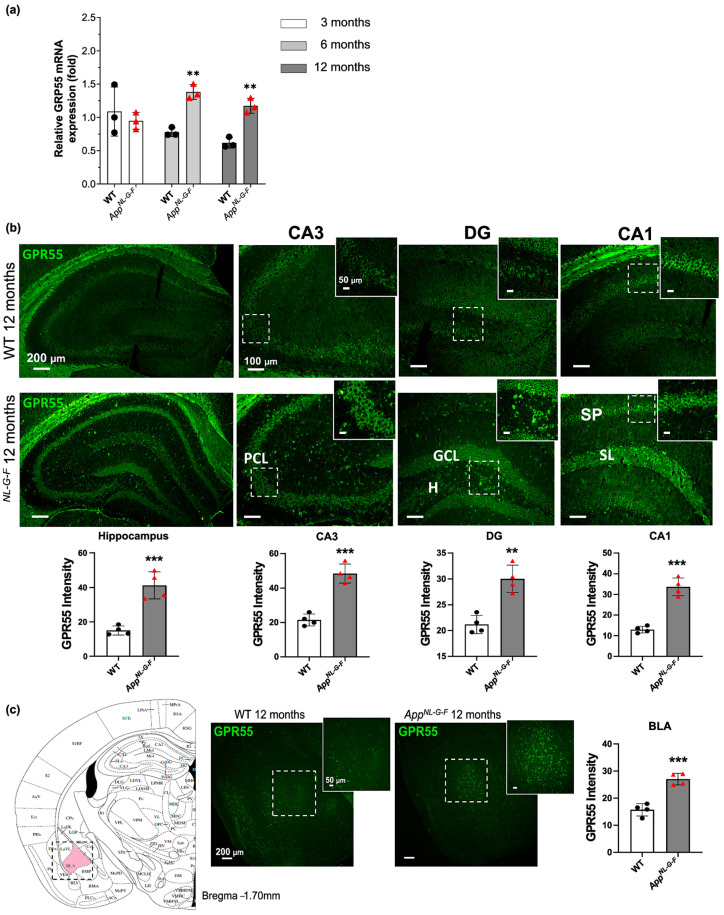
GPR55 mRNA expression and immunoreactivity in *App^NL-G-F^* and healthy control WT mice. (**a**) Relative GPR55 mRNA expression (fold) in WT and *App^NL-G-F^* mice at 3, 6, and 12 months old. Data represent mean ± S.E.M. One-way ANOVA was used, ** denotes *p* < 0.01. (**b**) Immunofluorescence images show the widespread distribution of GPR55 receptors (green) in the hippocampus. The immunoreactivity was seen more specifically in the pyramidal cell layer (PCL) of the CA3, in the granular cell layer (GCL) and the hilus (H) of the dentate gyrus (DG), and the stratum pyramidal (SP) and the stratum lacunose (SL) of the CA1. (**c**) Immunofluorescence images show the expression of GPR55 in the basolateral amygdala (BLA). Scale bars represent 200, 100 and 50 μm. Data represent mean ± S.E.M. (*n* = 4 mice per group and 2 sections per mouse were used for quantification). *t*-test was used, ** denotes *p* < 0.01 and *** denotes *p* < 0.001.

**Figure 4 biology-12-00805-f004:**
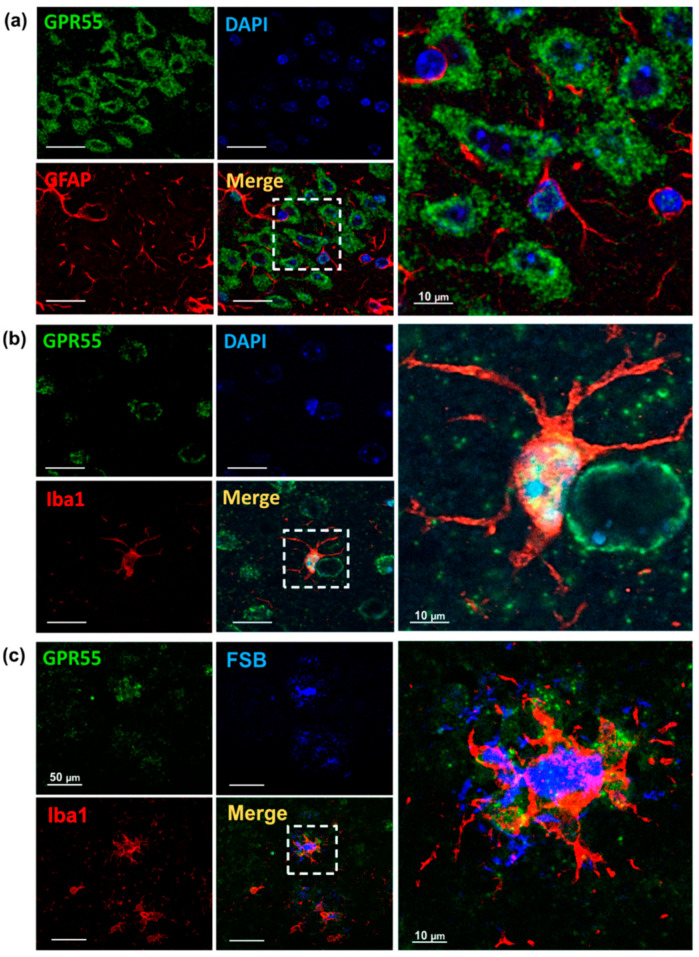
Astrocytes and microglia GPR55 immunoreactivity in 12-month-old *App^NL-G-F^* mice. (**a**) Representative immunostaining reveals a lack of overlapping staining between astrocytes marker (GFAP, red) and GPR55 (green) in a 12-month-old *App^NL-G-F^* mouse. (**b**) Representative images showing overlapping staining observed between microglia marker (Iba1, red) and GPR55 receptor (green). (**c**) Marked-activated microglia revealed by Iba1 (red) were found around amyloid plaques (FSB, blue). Abbreviations: GFAP: Glial fibrillary acidic protein, FSB: 1-Fluoro-2,5-bis(3-carboxy-4-hydroxystyryl) benzene, DAPI: (4′,6-diamidino-2-phenylindole).

**Figure 5 biology-12-00805-f005:**
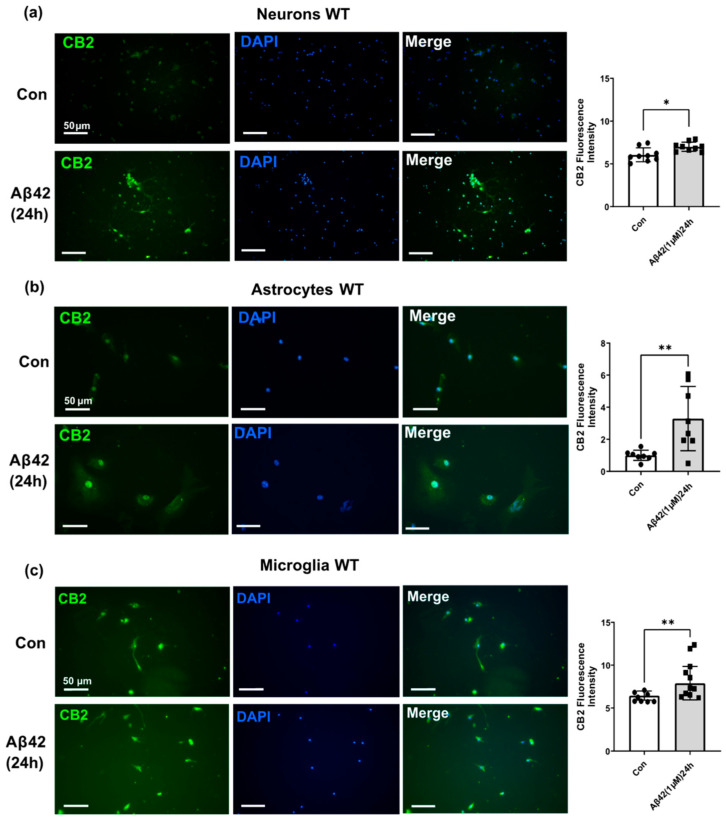
Effect of Aβ42 on CB2 expression in neurons, microglia, and astrocytes primary cell cultures from wild-type mice. Representative immunofluorescence images revealing CB2 (green) and DAPI (blue) immunostaining in (**a**) the neurons, (**b**) the astrocytes and, (**c**) microglia. Scale bars = 50 μm. Data represents changes in CB2 fluorescence intensity after 24 h Aβ42 1 μM treatment versus control non-treated (con). (*n* = 3 per group, at least 2 sections were used for quantification). Graphs are presented as Means ± SEM. *t*-test was used, * denotes *p* < 0.05 and ** denotes *p* < 0.01.

**Figure 6 biology-12-00805-f006:**
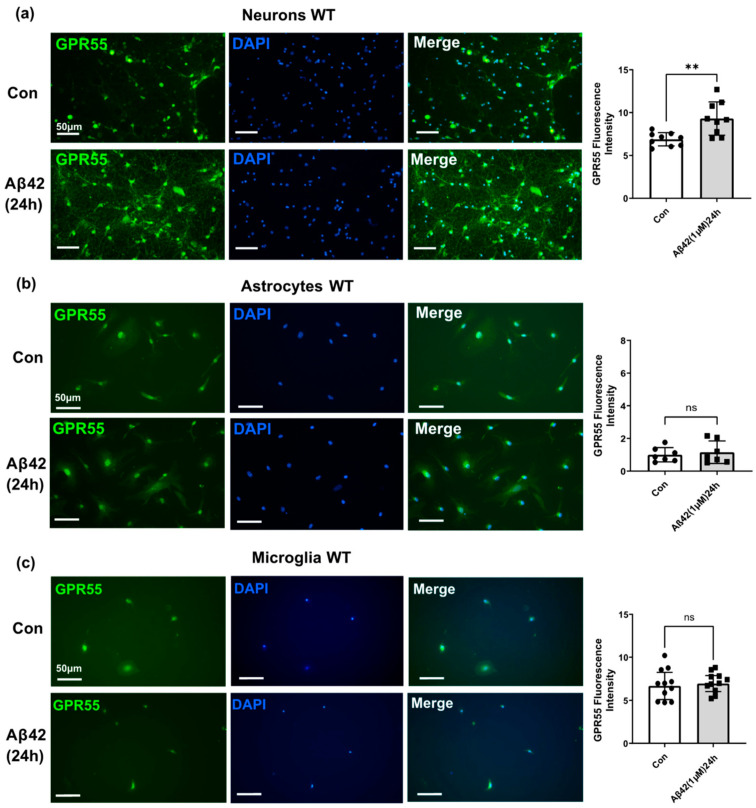
Effect of Aβ42 on GPR55 expression in neurons, microglia, and astrocytes primary cell cultures from wild-type mice. Representative immunofluorescence images immunostaining revealing GPR55 (green) and DAPI (blue) immunostaining in within (**a**) the neurons, (**b**) the astrocytes and, (**c**) microglia. Scale bars = 50 μm. Data represents changes in GPR55 fluorescence intensity after 24 h Aβ42 1 μM treatment versus control non-treated (con). (*n* = 3 per group, at least 2 sections were used for quantification). Graphs are presented as Means ± SEM. *t*-test was used, ** denotes *p* < 0.01 and ns mean no significant change.

**Figure 7 biology-12-00805-f007:**
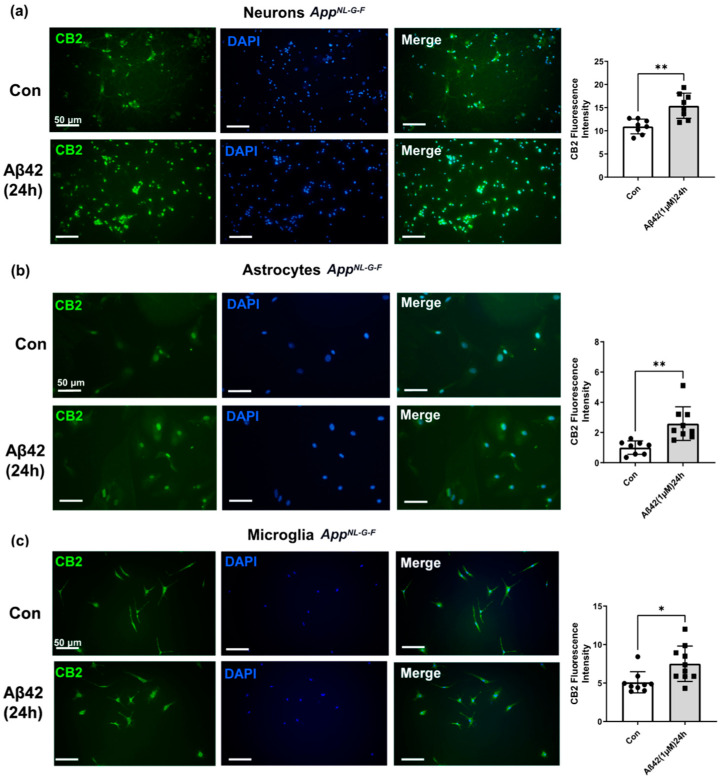
Effect of Aβ42 on CB2 expression in neurons, microglia, and astrocytes primary cell cultures from *App^NL-G-F^* mice. Representative immunofluorescence images revealing CB2 (green) and DAPI (blue) immunostaining in (**a**) the neurons, (**b**) the astrocytes and, (**c**) microglia. Scale bars = 50 μm. Data represents changes in CB2 fluorescence intensity after 24 h Aβ42 1 μM treatment versus control non-treated (con). (*n* = 3 per group, at least 2 sections were used for quantification). Graphs are presented as Means ± SEM. *t*-test was used, * denotes *p* < 0.05 and ** denotes *p* < 0.01.

**Figure 8 biology-12-00805-f008:**
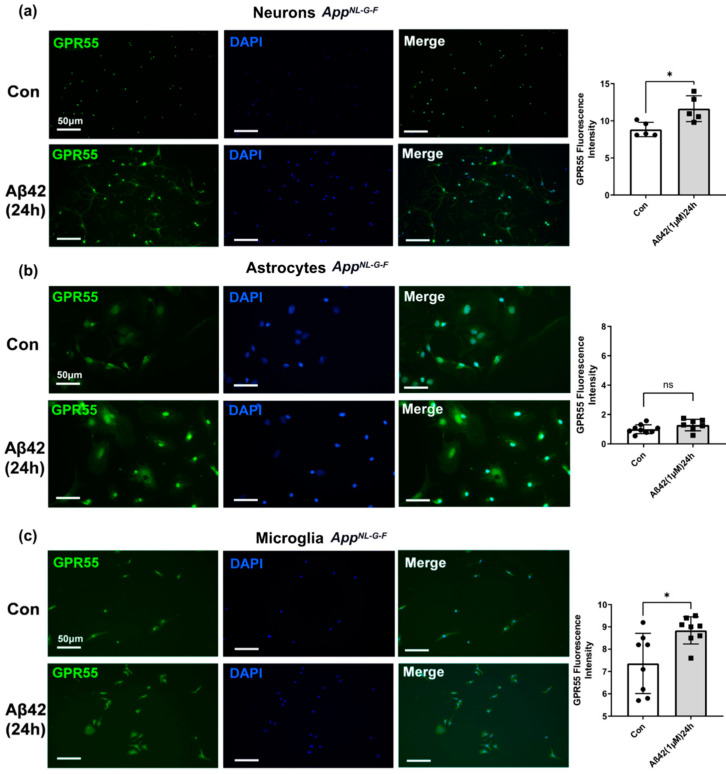
Effect of Aβ42 on GPR55 expression in neurons, microglia, and astrocytes primary cell cultures from *App^NL-G-F^* mice. Representative immunofluorescence images revealing GPR55 (green) and DAPI (blue) immunostaining in (**a**) the neurons, (**b**) the astrocytes and, (**c**) microglia. Scale bars = 50 μm. Data represents changes in GPR55 fluorescence intensity after 24 h Aβ42 1 μM treatment versus control non-treated (con). (*n* = 3 per group, at least 2 sections were used for quantification). Graphs are presented as Means ± SEM. *t*-test was used, * denotes *p* < 0.05 and ns means no significant change.

## Data Availability

The data presented in this study are available on request from the corresponding author.

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
