# Peer review of "The Expression of the Endocannabinoid Receptors CB2 and GPR55 Is Highly Increased during the Progression of Alzheimer’s Disease in AppNL-G-F Knock-In Mice"

_biology, 2023, doi:10.3390/biology12060805_

Round 1

Reviewer 1 Report

Dear Authors,

Minor considerations:

1) I suggest some minor corrections in the edition of the manuscript (for example character spacing is not consistent in the hole manuscript).

2) "... double immunofluorescence staining using selective astrocytes (GFAP), microglial (Iba1), and amyloid plaques (FSB) markers was performed." I suggest to modify this sentence in order to avoid misleading readers to think that could be a GFAP+Iba1 immunostaining.

3) Because no Pearson correlation coefficient nor Mander's overlap coefficient were used to quantify the degree of colocalization between markers, colocalization term in text should be avoided.

4) Legend on figure 3 describing GPR55 immunostaining in the basolateral amygdala is labeled as 'b' when it should be 'c'.

Major considerations:

1) After CB2 immunostaining, authors wrote "... in control mice (WT) positive CB2 staining was observed mainly in the neuronal cells in hilus of DG, whereas in AppNL-G-F mice, CB2 staining was observed not only in neurons but also in loci AB plaques (Figure 1b, white arrows)". How the authors support the following statments: (1) ABeta plaques are observed (identified with white arrows), and (2) that CB2 is observed in neurons.

2) Expression levels of CB2 and GPR55 were measured in vitro, using primary neuronal, astroglial and microglial primary cultures. Despite that probably western blot could be a more accurate technique to measure protein expression levels, images presented in Figures 5, 6 and 7 (with exception of 7c), show CB2 or GPR55 nuclear localization. However, as described in previous works, and also stated in the discussion section, both proteins are membrane receptors. If this was not an autofluorescence problem, authors should explain why both proteins are observed in the nuclei.

Kind regards,

Author Response

We would like to thank the reviewer for all the important observations.

Minor considerations: 

  • I suggest some minor corrections in the edition of the manuscript (for example character spacing is not consistent in the hole manuscript).  

Thank you very much for this observation, we have now gone through the manuscript and made the indicated corrections.

  • "... double immunofluorescence staining using selective astrocytes (GFAP), microglial (Iba1), and amyloid plaques (FSB) markers was performed." I suggest to modify this sentence in order to avoid misleading readers to think that could be a GFAP+Iba1 immunostaining.

            We have now changed the sentence as the reviewer suggested.

  • Because no Pearson correlation coefficient nor Mander's overlap coefficient were used to quantify the degree of colocalization between markers, colocalization term in text should be avoided. 

As suggested by the review now in the manuscript, we have substituted the word “colocalization” with “close proximity” and “overlapping staining”

  • Legend on figure 3 describing GPR55 immunostaining in the basolateral amygdala is labeled as 'b' when it should be 'c'.

We have now corrected the legend.

Major considerations: 

1) After CB2 immunostaining, authors wrote "... in control mice (WT) positive CB2 staining was observed mainly in the neuronal cells in hilus of DG, whereas in AppNL-G-F mice, CB2 staining was observed not only in neurons but also in loci Ab plaques (Figure 1b, white arrows)". How the authors support the following statments: (1) ABeta plaques are observed (identified with white arrows), and (2) that CB2 is observed in neurons.  

We apologize for the confusion, and the text has now been clarified and changed accordingly so that the text states that CB2 staining is found in the hippocampus and in granular aggregates. The identification of CB2 in amyloid plaques is presented in Figure 2.

2) Expression levels of CB2 and GPR55 were measured in vitro, using primary neuronal, astroglial and microglial primary cultures. Despite that probably western blot could be a more accurate technique to measure protein expression levels, images presented in Figures 5, 6 and 7 (with exception of 7c), show CB2 or GPR55 nuclear localization. However, as described in previous works and also stated in the discussion section, both proteins are membrane receptors. If this was not an autofluorescence problem, the authors should explain why both proteins are observed in the nuclei. 

CB2 is a G protein-coupled receptor that is primarily expressed on the cell membrane. However, some studies have reported the presence of intracellular CB2 receptors, including those localized within the endoplasmic reticulum and Golgi apparatus. Other labs have also described the presence of CB2-positive staining in cell nuclei as Jean-Ha Baek, et l., Acta Oto-Laryngologica, 128:9, 961-967, DOI: 10.1080/00016480701796944 (from page 926..” Positive immunoreactivity to CB2 was expressed as brown staining in the cytoplasm, nucleus, nuclear membrane and cell membrane.”..)

Similarly, GPR55 is mainly expressed in the cell membrane. However, there have been studies investigating the subcellular localization of the GPR55 receptor, and some of them have found evidence that it is also expressed intracellularly and/or in the nucleus. As:   Andradas et al., Oncogene, 30(2), 245-252. DOI: 10.1038/onc.2010.402; Lauckner et al., Proceedings of the National Academy of Sciences, 105(7), 2699-2704. DOI: 10.1073/pnas.0711278105; Ryberg et al., British Journal of Pharmacology, 152(7), 1092-1101. DOI: 10.1038/sj.bjp.0707460

However, considering all factors, we cannot exclude possible unspecific staining attributable to the antibody.

Reviewer 2 Report

Interesting paper; I hope the authors will proceed with the project to check the GPR55 and/or CB2 agonists on the Ab42 levels or even the level of the proinflammatory cytokines. It’d be interesting to see.

Method of calculation of the CB2 and GPR55 fluorescence intensity; 5/18 cells from three experiments?? Can we get some more explanation regarding that?
Have the authors counted five cells from two sections? Why astrocytes n=5 in supplementary while others are more?

Have the authors compared the GPR55 fluorescence in neurons between wt-control and APP-control? It seems in the Supplementary Figure 2 that it increased.

Why 1uM was chosen for the Ab42?

The purity of the cell cultures should be confirmed with cell-specific staining

Page 5, Line 211 typo: ‘detatched’

Page 9, line 309: CB2-activated microglia ???

Page 11 Line 349; after the bracket,  is that the end of the sentence?

Page 11, Line 356 -the sentence is not finished.

Page 12, Lines 368-370 -those sentences don’t make sense.

No need to put ‘t-test’ everywhere in the results, it’s enough to be put in the method and saying, statistic were made in t-test unless otherwise specified
Page 14, line 414; Figure 8a nad Figure 8c, respectively

Page 15, Line 418 – check the grammar please

Page 17, Line 523 – “transition of microglial in the activated state”?

Author Response

Interesting paper; I hope the authors will proceed with the project to check the GPR55 and/or CB2 agonists on the Ab42 levels or even the level of the proinflammatory cytokines. It’d be interesting to see. 

We really appreciate the encouraging positive feedback from the reviewer.

Method of calculation of the CB2 and GPR55 fluorescence intensity; 5/18 cells from three experiments?? Can we get some more explanation regarding that? Have the authors counted five cells from two sections? Why are astrocytes n=5 in supplementary while others are more? 

We apologize for the unclear description in the previous version of the manuscript, now we have changed it. The immunofluorescence quantification of primary cell culture was performed at least in triplicates, and from each mounted slide, at least 2 images were used for quantification. However, in the GPR55 staining of neuronal group AppNL-G-F treated with Ab42 (Figure 8), one data point was identified and excluded as an outlier; hence, 5 data points are presented in the graph (we believe the reviewer is referring to these data when mentioning astrocytes since the n in the astrocyte groups are 7 or higher).

Have the authors compared the GPR55 fluorescence in neurons between wt-control and APP-control? It seems in Supplementary Figure 2 that it increased. 

Yes, it was compared, but not significant (Now Supplementary Figure 4).

Why 1µM was chosen for the Aβ42? 

In our lab, we have several years of experience in producing (in collaboration with Dr. Gefei Chen, Co-Author in the paper) and testing Aβ42 in cell models.  Our research utilizing murine primary cell cultures has demonstrated that, upon administering a dose of 1 µM Aβ42 for a 24-hour duration, we successfully and consistently replicated Alzheimer's disease-like Aβ42 neurotoxicity and elicited microglial and astroglia activation (https://doi.org/10.1101/2022.10.24.513473).

Furthermore, other laboratories have employed Aβ42 at concentrations of 1 µM to achieve analogous objectives. A few representative examples include the following studies:

Beretta et al., Sci Rep. 2020 Nov 12;10(1):19656. doi: 10.1038/s41598-020-72355-2.

Mastroeni et al., PLoS One. 2013;8(1):e53349. doi: 10.1371/journal.pone.0053349.

Bernabeu-Zornoza et al., Int J Mol Sci. 2021;22(17):9537. doi: 10.3390/ijms22179537.

The purity of the cell cultures should be confirmed with cell-specific staining.

The protocol used in our study is a well-established protocol that was described in detail by Saura et al. in 2003 (Glia. 2003 Dec;44(3):183-9. doi: 10.1002/glia.10274) and also used by several other groups, for example by Ayata et al., 2018 (Nat Neurosci. 2018 Aug;21(8):1049-1060. doi: 10.1038/s41593-018-0192-3). With this protocol, confluent mixed glial cultures are subjected to mild trypsinization, which detaches the layer containing virtually all the astrocytes, leaving undisturbed a population of firmly attached cells identified as >98% microglia. The detached astrocytes are then cultured separately. An example of such an experiment in which we stain for GFAP and Iba1 to determine the cell identity is presented in a new supp. Figure 2.

We appreciate the following referee’s comments and suggestions. We have corrected all the below points as a following:

Page 5, Line 211 typo: ‘detatched’ now changed to ´detached` (now 236)

Page 9, line 309: CB2-activated microglia ??? now changed to `revealed positive CB2 staining within activated microglia in the proximity of…` (now 347).

Page 11 Line 349; after the bracket,  is that the end of the sentence? Sorry for the inconvenience; we have fixed it (now 395).

Page 11, Line 356 -the sentence is not finished. We have now fixed the sentence (now 402).

Page 12, Lines 368-370 -those sentences don’t make sense. We have now fixed the sentence (now 420 421).

No need to put ‘t-test’ everywhere in the results, it’s enough to be put in the method and saying, statistics were made in t-test unless otherwise specified. As suggested by the reviewer, we have now fixed this part and kept the t-test only in the methods.

Page 14, line 414; Figure 8a and Figure 8crespectively  We have now fixed it (now 488).Page 15, Line 418 – check the grammar please. We have now fixed the grammar of this sentence (now 492).

Page 17, Line 523 – “transition of microglial in the activated state”? We apologize for the unclear sentence. We have changed it now, (now 609 610).

Reviewer 3 Report

The expression of the endocannabinoid receptors CB2 and GPR55 is highly increased during the progression of Alzheimer´s disease in AppNL-G-Fknock-in mice

The present study investigates the expression levels of the cannabinoid/lysophospholipid receptors CB2 and GPR55 at different stages of AD. The study demonstrates the co-localization of CB2 and GPR55 with Aβ-plaque in AD mouse model and analyses the pathophysiological mechanism in neuronal and glial primary cell cultures. The approach and the overall design of the study are good. Thus, the findings will help potential readers to understand the mechanisms underlying the different neurobiological brain alterations during AD pathologies, including the endocannabinoid system (ECS) and associated lipid transmitter-based signaling systems.

Author Response

The expression of the endocannabinoid receptors CB2 and GPR55 is highly increased during the progression of Alzheimer´s disease in AppNL-G-Fknock-in mice 

The present study investigates the expression levels of the cannabinoid/lysophospholipid receptors CB2 and GPR55 at different stages of AD. The study demonstrates the co-localization of CB2 and GPR55 with Aβ-plaque in AD mouse model and analyses the pathophysiological mechanism in neuronal and glial primary cell cultures. The approach and the overall design of the study are good. Thus, the findings will help potential readers to understand the mechanisms underlying the different neurobiological brain alterations during AD pathologies, including the endocannabinoid system (ECS) and associated lipid transmitter-based signaling systems. 

We thank this reviewer for appreciating our study.

Round 2

Reviewer 1 Report

Dear Author,

Thanks for accepting the suggested modifications. This is a really interesting project. It would be nice if the group continues exploring alterations of the endocannabinoid system in neurodegeneration. My suggestion for authors is to provide the control images used for the immunostaining experiment. If authors does not have control images, I recommend to discard image 8 from the manuscript. If authors believe this experiment is fundamental for the paper, my recommendation is to confirm that observation using subcellular fragmentation experiments following by western blot to prove that images from figure 8 are right. Also, nuclear localization of GPR55 should be included in the discussion.

Author Response

Thanks for accepting the suggested modifications. This is a really interesting project. It would be nice if the group continues exploring alterations of the endocannabinoid system in neurodegeneration.

We really appreciate the Review's encouragement in continuing our studies of the endocannabinoid system characterization in AD pathology.

My suggestion for authors is to provide the control images used for the immunostaining experiment. If authors does not have control images, I recommend to discard image 8 from the manuscript. If authors believe this experiment is fundamental for the paper, my recommendation is to confirm that observation using subcellular fragmentation experiments following by western blot to prove that images from Figure 8 are right. Also, nuclear localization of GPR55 should be included in the discussion.

We appreciate the feedback provided by the reviewer. The control images from Figure 8, omitting GPR55 primary antibody (no anti-GPR55) with the only secondary antibody anti-rabbit, are now available as Supplementary Figure 3. In agreement with the reviewer's comment on the nuclear localization of GPR55, we have now included in the discussion a new paragraph related to the positive nuclear staining observed for GPR55 and CB2 (page 20).

Furthermore, the recommendation to validate the GPR55 and CB2 nuclear presence with subcellular fragmentation experiments, followed by Western blot, is a valuable suggestion. We agree that this approach could further confirm a nuclear expression of GPR55 and CB2; however, we think the primary aim of our current manuscript is focused on the evaluation of the levels, rather than the intracellular localization, of CB2 and GPR55 gene and protein expression across different stages of Alzheimer's Disease in the AppNL-G-F mouse model. However, we are aware of the importance of this aspect and recognize the Reviewer's suggestion as a valuable direction for our future research. 

Round 3

Reviewer 1 Report

Dear Author,

Thanks for the positive reply to the previous suggestions. Because images showed in Figure S3 are showing unspecific labeling from the secondary antibody, I’m sorry but I believe that specific result loses reliability with that control.

Again, If authors have samples for western blot, I recommend to measure the expression of proteins  by western blot, since cellular localization is not the focus of the present work, it would be better and cleaner with western blot than with immunocytochemistry.

Kind regards,

Author Response

Answers to the Reviewer # 1

Thanks for the positive reply to the previous suggestions. Because images showed in Figure S3 are showing unspecific labeling from the secondary antibody, I’m sorry but I believe that specific result loses reliability with that control. 

Thank you for your comments and the important suggestion. However, to our knowledge, we think the negative controls are good. The presence of some background staining from the secondary antibody is something very common in immunofluorescence analysis. As researchers with several years of experience, we think that is quite a common event. We purposely included areas with a few staining to show that the secondary antibody was present even though the large majority of cells were not stained. From our side, it will be more problematic to judge a completely black picture with no signal supporting the presence of secondary antibodies.  

In general, we think that the negative controls shown in Figure S3 support a true staining of the cells in Figure 8. 

Again, If authors have samples for western blot, I recommend to measure the expression of proteins by western blot, since cellular localization is not the focus of the present work, it would be better and cleaner with western blot than with immunocytochemistry. 

Unfortunately, and we are sorry for that, right now, we do not have any tissues in our hands available.